# Clinical Analysis of Perioperative Outcomes on Neoadjuvant Hormone Therapy before Laparoscopic and Robot-Assisted Surgery for Localized High-Risk Prostate Cancer in a Chinese Cohort

**Guangyu Sun †, Zhengxin Liang †, Yuchen Jiang, Shenfei Ma, Shuaiqi Chen and Ranlu Liu ***

Department of Urology, Tianjin Institute of Urology, The Second Hospital of Tianjin Medical University,
Tianjin 300211, China
* Correspondence: ranluliu@126.com
† These authors contributed equally to this work.

**Abstract:** Objective: To analyze the perioperative outcomes of neoadjuvant hormone therapy (NHT) before laparoscopic and robot-assisted surgery for localized high-risk prostate cancer in a Chinese cohort. Methods: The clinical data of 385 patients with localized high-risk prostate cancer who underwent radical prostatectomy (RP) in our hospital from January 2019 to June 2021 were analyzed retrospectively, including 168 patients with preoperative NHT and 217 patients with simple surgery. Clinical characteristics were compared in the above two groups, the laparoscopic RP (LRP) cohort (n = 234) and the robot-assisted laparoscopic radical prostatectomy (RALP) cohort (n = 151), respectively. Results: In the overall cohort, compared with the control group, the NHT group had a shorter operative time, less blood loss, a lower positive surgical margin rate, and a higher proportion of Gleason score (GS) downgrading after the operation ($p < 0.05$). However, there was no significant difference in hospitalization time, biochemical recurrence, urine leakage, urinary continence, or prostate-specific antigen (PSA) progression-free survival ($p > 0.05$). In the LRP cohort, it was found that the NHT group also had shorter operative time, less blood loss, lower positive surgical margin rate, a higher proportion of GS downgrading after the operation, and faster recovery of urinary control than the control group ($p < 0.05$). There was no marked difference in hospitalization time, biochemical recurrence, urinary leakage, or PSA progression-free survival. However, in the RALP cohort, the NHT group had a significant difference in the GS downgrading after the operation compared with the control group ($p < 0.05$). In the overall cohort, multiple analyses showed that initial PSA level, GS at biopsy, clinical T stage, lymph node invasion, use of NHT, and surgical methods were significantly associated with positive surgical margin ($p < 0.05$) while NHT did not account for biochemical recurrence ($p > 0.05$). Conclusions: NHT can lower the difficulty of surgery, reduce positive surgical margin rate, and help recovery in short-term urinary control in patients with high-risk prostate cancer after LRP. However, we do not have evidence on the benefit of NHT in high-risk PCa patients treated with RALP. For these patients, surgery can be performed as early as possible.

**Keywords:** prostate cancer; neoadjuvant hormone therapy; laparoscopic radical prostatectomy; robot-assisted radical prostatectomy

## 1. Introduction

Prostate cancer has the highest incidence and ranks second place in mortality among male cancers [1]. Due to the absence of early symptoms and sufficient prostate-specific antigen (PSA) screening, approximately 70% of patients in China are initially diagnosed with high-risk or even advanced prostate cancer [2,3]. Although each guideline differs slightly, comprehensive treatments are usually recommended, including surgery, radiation

therapy, or new endocrine therapeutic drugs [4]. With the advantage of excellent control of local primary tumors, availability of accurate pathological staging, and the following guidance for adjuvant therapy, radical prostatectomy (RP) becomes one of the most effective treatments for low- and intermediate-risk prostate cancer and the main option for localized high-risk prostate cancer [5]. The European Association of Urology guidelines and the National Comprehensive Cancer Network guidelines also suggest that surgery should be one of the comprehensive treatments for high-risk prostate cancer [4]. However, in high-risk prostate cancer, pathologic findings reveal a relatively high proportion of positive surgical margins, suggesting that surgery may not be curative and therefore adjuvant therapy is necessary [6]. However, the clinical findings of neoadjuvant hormone therapy (NHT) before radical prostatectomy are inconsistent among various studies. Some studies have found that neoadjuvant hormone therapy prior to RP helps reduce the incidence of positive surgical margins and lymph node infiltration [7,8], but others have suggested that neoadjuvant hormone therapy prior to RP does not have a significant survival benefit for patients, including overall survival and disease-free survival [9], and therefore preoperative neoadjuvant therapy is not specifically recommended by current guidelines for patients with high-risk prostate cancer.

Currently, robotic-assisted radical prostatectomy (RALP) has become the main modality in prostate cancer surgery [10], and it has been reported that more than 85% of RP procedures in the United States are RALP [11]. Robotic-assisted systems have significant advantages over laparoscopic radical surgery in terms of improved ergonomics, a three-dimensional magnified view, and tremor filtration, with lower rates of urinary incontinence and biochemical recurrence [12]. However, the survival benefit of robotic-assisted radical surgery for patients with high-risk prostate cancer is not clear, especially in some patients who have undergone neoadjuvant hormonal therapy [13]. Our study retrospectively analyzed the perioperative outcomes of NHT in patients with high-risk prostate cancer and explored the benefits of NHT for patients with two different surgical approaches, RALP and LRP.

## 2. Materials and Methods

### 2.1. Patients and Treatments

The clinical data of 385 patients with localized high-risk prostate cancer treated in the second hospital of Tianjin Medical University from January 2019 to June 2021 were collected. Inclusion criteria: (1) localized high-risk prostate cancer diagnosed by pathology and imaging; (2) NHT cohort: NHT > 3 months; (3) LRP or RALP. Exclusion criteria: (1) incomplete relative data; (2) patients with metastatic lesions. High-risk prostate cancer was defined as clinical stage T3-4 and/or PSA > 20 ng/mL and/or Gleason score (GS) 8–10.

NHT includes oral administration of bicalutamide 50 mg once a day plus goserelin 3.6 mg or leuprorelin 3.75 mg subcutaneously or triptorelin 3.75 mg intramuscularly once a month. LRP or RALP were carried out by the same group of urologists.

All included cases underwent preoperative pelvic magnetic resonance imaging (MRI), thoracoabdominal CT and bone scan (ECT) to determine TNM staging (2017, AJCC). Pathological histological classification was performed using the Gleason scoring system (2016, WHO) for prostate cancer. All patients with adverse pathology received routine adjuvant hormone therapy ± external beam radiotherapy after surgery. Regular postoperative inpatient or outpatient reviews were performed regularly, with PSA every 1–3 months and ultrasound, MRI, and bone scan every 6–12 months, with a follow-up period of 12–36 months and a median follow-up of 25 months.

### 2.2. Perioperative Outcomes

The perioperative outcomes were observed and recorded, including operation time, intraoperative bleeding, postoperative urinary leakage, rate of positive surgical margin, GS downgrading, length of hospital stay, recovery of urinary control, biochemical recurrence, and PSA progression-free survival. Biochemical recurrence was determined using the

European Urological (EAU) criteria of two consecutive serum PSA levels > 0.2 ng/mL. Progression-free survival was defined as time from prostatectomy to biochemical recurrence.

### 2.3. Statistical Analysis

SPSS for Windows (Version 25.0) was used to conduct a statistical analysis of the data. The Kolmogorov–Smirnov test was used to judge the normal distribution of measurement data. The measurement data with normal distribution were expressed as mean ± standard deviation. Otherwise, the median ± interquartile range was used. The enumeration data were expressed as the number of cases and corresponding percentages. *t*-test or analysis of variance was used for component comparison of measurement data, and chi-square test or Fisher exact test were used for comparison between count data groups. Graphpad prism was used to analyze the PSA progression-free survival. Univariate and multiple logistic regression analyses were used to determine risk factors for positive surgical margins and biochemical recurrence. $p < 0.05$ was considered statistically significant.

### 3. Results

*3.1. Comparison of General Clinical Characteristics and Perioperative Characteristics between the NHT and Control Group in the Overall Cohort*

A total of 385 patients with localized high-risk prostate cancer were included in the overall cohort, including 234 cases treated with LRP and 151 cases treated with RALP. In the LRP cohort, 100 cases received NHT before operation and 134 cases were treated only by operation. In the RALP cohort, 68 cases received NHT before operation and 83 cases received simple surgical treatment.

In the overall cohort, there was no difference between the NHT group (168 patients) and the control group (217 patients) in age, BMI, PSA, prostate volume, Gleason score, T stage, or surgical methods ($p > 0.05$, Table 1). Compared with the control group, the NHT group had significantly shorter operative time, less blood loss, a lower positive surgical margin rate (28.6% vs. 38.3%), and a higher proportion of GS downgrading after the operation (24.4% vs. 13.8%), while there was no statistical significance in leakage rate (6% vs. 7.4%), biochemical recurrence rate (22.6% vs. 25.8%), or recovery time of urinary control (Table 2). No significant difference was found in PSA progression-free survival between the two groups (Table 2).

**Table 1.** Comparison of general clinical characteristics between the NHT and control groups in the overall cohort.

| | NHT Group (n = 168) | Control Group (n = 217) | *p*-Value |
|---|---|---|---|
| Age (year-old) | 67 ± 9 | 68 ± 9 | 0.100 |
| BMI (kg/m$^2$) | 22.45 ± 3.14 | 23.05 ± 2.92 | 0.057 |
| Volume (mL) | 39.33 ± 22.97 | 38.02 ± 24.53 | 0.261 |
| Initial PSA (ng/mL) | 23 ± 23.84 | 20 ± 12.26 | 0.075 |
| GS at biopsy | | | 0.917 |
| GS = 7 | 8 (4.8%) | 8 (3.6%) | |
| GS = 8 | 89 (53%) | 118 (54.4%) | |
| GS = 9 | 53 (31.5%) | 65 (30%) | |
| GS = 10 | 18 (10.7%) | 26 (12%) | |
| Initial T stage | | | 0.705 |
| cT3 | 158 (94%) | 206 (94.9%) | |
| cT4 | 10 (6%) | 11 (5.1%) | |
| Lymph node invasion | | | 0.560 |
| $N_0$ | 127 (75.6%) | 158 (72.8%) | |
| $N_X$ | 41 (24.4%) | 59 (27.2%) | |
| surgical method | | | 0.657 |
| LRP | 100 (59.5%) | 134 (61.8%) | |
| RRP | 68 (40.5%) | 83 (38.2%) | |

**Table 2.** Comparison of postoperative clinical characteristics between the NHT and control groups in overall cohort.

| | NHT Group (n = 168) | Control Group (n = 217) | *p*-Value |
|---|---|---|---|
| Operative time (minutes) | 108.99 ± 22.74 | 118.55 ± 24.71 | 0.007 |
| Blood loss (mL) | 110.76 ± 45.67 | 138.20 ± 48.17 | <0.001 |
| Urine leakage | 10 (6%) | 16 (7.4%) | 0.582 |
| Positive surgical margin | 48 (28.6%) | 83 (38.3%) | 0.02 |
| GS decreased after operation | 41 (24.4%) | 30 (13.8%) | 0.008 |
| Hospitalization (days) | 7 ± 2 | 7 ± 2 | 0.086 |
| Follow up time (months) | 24 ± 7 | 25 ± 5 | 0.48 |
| urinary continence | | | 0.06 |
| <1 month | 101 (60.1%) | 97 (44.7%) | |
| 1–3 month | 50 (29.8%) | 98 (45.2%) | |
| >3 month | 17 (10.1%) | 22 (10.1%) | |
| BCR | 38 (22.6%) | 56 (25.8%) | 0.47 |
| PSA progression free survival (months) | 23.5 ± 7 | 24 ± 5 | 0.152 |

*3.2. Comparison of Postoperative Clinical Characteristics between the NHT and Control Groups in the LRP Cohort*

In the LRP cohort, the same statistically significant trend was observed in the NHT group, including shorter operative time, less blood loss, lower positive surgical margin rate (34.00% vs. 49.3%), a higher proportion of GS decreasing after the operation (25.0% vs. 13.4%), and shorter recovery time of urinary control ($p < 0.05$, Table 3). There is no statistical significance in biochemical recurrence rate (24% vs. 29.9%), the leakage rate (7.0% vs. 9.0%) ($p = 0.26$, Table 3), or PSA progression-free survival ($p = 0.153$, Table 3).

**Table 3.** Comparison of postoperative clinical characteristics between the NHT and control groups in the LRP cohort.

| | NHT Group (n = 100) | Control Group (n = 134) | *p*-Value |
|---|---|---|---|
| Operative time (minutes) | 114 ± 29 | 128 ± 31 | <0.001 |
| Blood loss (mL) | 112.65 ± 68.77 | 151.5 ± 33.8 | <0.001 |
| Urine leakage | 7 (7%) | 12 (9%) | 0.26 |
| Positive surgical margin | 34 (34%) | 66 (49.3%) | 0.588 |
| GS decreased after operation | 25 (25%) | 18 (13.4%) | 0.024 |
| Hospitalization (days) | 7 ± 1 | 7 ± 1 | 0.957 |
| Follow up time (months) | 24 ± 8 | 25 ± 4 | 0.056 |
| urinary continence | | | 0.02 |
| <1 month | 58 (58%) | 53 (39.6%) | |
| 1–3 month | 34 (34%) | 65 (48.5%) | |
| >3 month | 8 (8%) | 16 (11.9%) | |
| BCR | 24 (24%) | 40 (29.9%) | 0.321 |
| PSA progression free survival (months) | 23 ± 8 | 24 ± 5 | 0.153 |

*3.3. Comparison of Postoperative Clinical Characteristics between the NHT and Control Groups in the RALP Cohort*

In the RALP cohort, rather than the marked difference in the proportion of GS downgrading ($p < 0.05$, Table 4), other perioperative outcomes including operative time, blood loss, positive surgical margin rate, urinary control, biochemical recurrence rate, and PSA progression-free survival were not significantly different between NHT group and control group ($p > 0.05$, Table 4).

**Table 4.** Comparison of postoperative clinical characteristics between the NHT and control groups in RALP cohort.

|  | NHT Group (n = 68) | Control Group (n = 83) | *p*-Value |
|---|---|---|---|
| Operative time (minutes) | 98.25 ± 18.6 | 100.5 ± 21.06 | 0.583 |
| Blood loss (mL) | 109.6 ± 19.62 | 105.15 ± 24.4 | 0.279 |
| Urine leakage | 3 (4.4%) | 12 (4.8%) | 0.906 |
| Positive surgical margin | 14 (20.6%) | 17 (20.5%) | 0.987 |
| GS decreased after operation | 20 (29.4%) | 12 (14.5%) | 0.025 |
| Hospitalization (days) | 5 ± 2 | 6 ± 2 | 0.208 |
| Follow up time (months) | 24 ± 6 | 25 ± 6 | 0.542 |
| urinary continence |  |  | 0.079 |
| <1 month | 43 (63.3%) | 44 (53%) |  |
| 1–3 month | 16 (23.5%) | 33 (39.8%) |  |
| >3 month | 9 (13.2%) | 6 (7.2%) |  |
| BCR | 14 (20.6%) | 16 (19.3%) | 0.84 |
| PSA progression free survival (months) | 24 ± 6 | 24 ± 6 | 0.592 |

*3.4. Univariate and Multiple Logistic Regression Analysis of Positive Surgical Margins and Biochemical Recurrence in Overall Cohort*

In the overall cohort, univariate analysis showed that volume of the prostate, initial PSA level, GS at biopsy, initial T stage, lymph node invasion, use of NHT and surgical method were significantly associated with positive surgical margins. Multiple analysis showed that initial PSA level, GS at biopsy, lymph node invasion, use of NHT, and surgical methods were significantly associated with positive surgical margin while NHT did not account for biochemical recurrence (Table 5).

**Table 5.** Univariate and multiple logistic regression analysis of positive surgical margins in the overall cohort.

|  | Univariate Analysis | | | Multiple Logistic Regression | | |
|---|---|---|---|---|---|---|
|  | OR | 95%CI | *p* | OR | 95%CI | *p* |
| Age (year-old) | 0.982 | 0.952–1.012 | 0.227 | - | - | - |
| BMI (kg/m$^2$) | 1.093 | 0.989–1.206 | 0.08 | - | - | - |
| Volume (mL) | 1.02 | 1.004–1.036 | 0.012 | 1.003 | 0.981–1.025 | 0.797 |
| Initial PSA (ng/mL) | 1.084 | 1.061–1.107 | <0.001 | 1.102 | 1.073–1.132 | <0.001 |
| GS at biopsy (GS7-8, GS9-10) | 12.626 | 3.873–41.158 | <0.001 | 5.220 | 2.688–10.134 | <0.001 |
| Initial T stage (T3, T4) | 6.933 | 1.333–6.194 | 0.019 | 2.671 | 0.536–13.312 | 0.231 |
| Lymph node invasion (N$_0$, N$_X$) | 16.856 | 9.558–29.728 | <0.001 | 5.443 | 2.378–12.462 | <0.001 |
| Treatment (Control, NHT) | 0.646 | 0.419–0.995 | 0.047 | 0.365 | 0.190–0.700 | 0.002 |
| surgical method (LRP, RALP) | 0.346 | 0.216–0.555 | <0.001 | 0.179 | 0.091–0.354 | <0.001 |

Univariate analysis showed that volume of the prostate, initial PSA level, GS at biopsy, initial T stage, positive surgical margin, and lymph node invasion were significantly associated with biochemical recurrence. Multiple analysis of risk factors for biochemical recurrence in the overall cohort showed that only initial PSA levels, positive surgical margins, and lymph node invasion were independent risk factors for biochemical recurrence. (Table 6).

**Table 6.** Univariate and multiple logistic regression analysis of biochemical recurrence in the overall cohort.

| | Univariate Analysis | | | Multiple Logistic Regression | | |
|---|---|---|---|---|---|---|
| | OR | 95%CI | *p* | OR | 95%CI | *p* |
| Age (year-old) | 1.033 | 0.998–1.07 | 0.068 | - | - | - |
| BMI (kg/m$^2$) | 1.091 | 0.978–1.216 | 0.118 | - | - | - |
| Volume (mL) | 1.034 | 1.016–1.053 | <0.001 | 1.021 | 0.995–1.048 | 0.112 |
| Initial PSA (ng/mL) | 1.069 | 1.05–1.089 | <0.001 | 1.027 | 1.020–1.066 | 0.018 |
| GS at biopsy (GS7-8, GS9-10) | 3.418 | 3.099–6.921 | 0.006 | 1.952 | 0.796–4.786 | 0.144 |
| Initial T stage (T3, T4) | 11.733 | 1.421–8.220 | <0.001 | 1.862 | 0.487–7.112 | 0.363 |
| Lymph node invasion ($N_0$, $N_X$) | 37.712 | 19.938–71.331 | <0.001 | 25.031 | 9.929–63.102 | <0.001 |
| Treatment (Control, NHT) | 0.84 | 0.524–1.348 | 0.471 | - | - | - |
| surgical method (LRP, RALP) | 0.659 | 0.403–1.078 | 0.096 | - | - | - |
| Positive surgical margin | 11.885 | 6.861–20.586 | <0.001 | 3.597 | 1.657–7.808 | <0.001 |

## 4. Discussion

Traditionally, conservative treatments such as radiotherapy, surgery, and hormone therapy are preferred for high-risk prostate cancer [14]. However, with the continuous improvement of medical technology and minimally invasive surgical techniques, RP is considered the prior treatment choice for patients with localized high-risk prostate cancer [15,16]. NHT is widely used for the initial treatment of high-risk prostate cancer. However, the role of preoperative NHT in high-risk prostate cancer is still controversial due to potential drawbacks such as ineffectiveness, delayed access to surgery, and increased surgical complications [17]. In a prospective study in which investigators matched patients treated with/without NHT on a 1:2 basis, recipients of NHT groups had lower rates of positive surgical margins, seminal vesicle infiltration, and extracapsular extension than non-subjects, in addition to a lower rate of perioperative complications (7.4% vs. 18.4%) [18]. A meta-analysis conducted by Shelley showed that NHT treatment contributed to improved adverse pathologic outcomes [19]. Joung found that 6 (5.4%) of 111 postoperative specimens of high-risk prostate cancer patients treated with NHT were free of tumor residuals [20]. Our study likewise found a significantly lower rate of positive surgical margin in the NHT group (28.6% vs. 38.3%) and a higher rate of GS downgrading (24.4% vs. 13.8%) than in the control group. Another retrospective study found a significant decrease in operative time, blood loss, and positive surgical margins in the NHT group [21], consistent with the results of our analysis.

NHT could shrink the prostate volume and facilitate the surgery, significantly shortening the operative time and reducing intraoperative blood loss. As the prostate is relatively deep and fixed, the operative time and intraoperative bleeding are closely related to the adhesions around the tumor, thus bleeding is extremely easy when dealing with bilateral ligaments and freeing the vas deferens seminal vesicles, especially for patients with locally advanced prostate cancer [22]. With preoperative NHT, intraoperative anatomical structure was identified much more easily and the surgical difficulty was relatively low [23].

There are inconsistent results from various studies on whether neoadjuvant hormone therapy improves patients' prognosis. A multicenter study analyzed high-risk prostate cancer-related mortality after NHT in 1573 patients and found that preoperative NHT significantly reduced postoperative prostate cancer-related mortality [24]. Berglund concluded that NHT treatment improved progression-free and overall survival [25]. On the other hand, Scorieri concluded that NHT treatment did not benefit the survival of prostate cancer patients [26]. Our study found that NHT reduced the rate of positive surgical margin but there was no significant benefit in the NHT group in terms of biochemical recurrence and PSA progression-free survival by late follow-up (*p* > 0.05).

Currently, robot-assisted surgery is gradually replacing LRP as the preferred choice for the surgical treatment of localized prostate cancer [13]. Similarly, RALP is safe for patients with high-risk prostate cancer and has some advantages over general laparoscopic

surgery in terms of operative time, hospital stay, and postoperative complication rate [27]. A meta-analysis by Ashutosh et al. related to the surgical approach to prostate cancer found lower positive surgical margins, less bleeding, less complications such as urinary incontinence and rectal injury for RALP compared to LRP and ORP, but no difference in biochemical recurrence [28], which is consistent with our findings. However, preoperative NHT is currently focused on laparoscopic surgery, and few studies have reported the clinical perioperative outcomes of ADT in robot-assisted surgery. However, side effects of neoadjuvant hormone therapy are also evident, most notably in cardiovascular risk. Neoadjuvant hormone therapy may increase the incidence of cardiovascular risk in patients, and therefore neoadjuvant hormone therapy for high-risk prostate cancer patients should be analyzed individually and risk stratified, which is subject to further study [29]. In this study, we subsequently performed a statistical analysis of the NHT group and the control group in the LRP cohort and the RALP cohort, respectively, and found that NHT was more efficient for laparoscopic surgery, with significant advantages in intraoperative bleeding, positive margin rate, GS downgrading, and recovery of urinary control ($p < 0.05$), consistent with the overall cohort. However, its effect on the RALP cohort was not that obvious. Therefore, we can conclude that the high accuracy of the robot-assisted system reduces the impact of prostate volume and stricture space during the procedure. we do not have evidence on the benefit of NHT in high-risk PCa patients treated with RALP. For these patients, surgery can be performed as early as possible.

Our study also has some limitations. First, our study is a retrospective study and inevitably suffers from selection bias, as the choice of neoadjuvant hormone therapy only included bicalutamide and not the latest new endocrine therapies such as enzalutamid, apalutamid and datolutamid. In addition, our surgical urethral reconstruction modalities are all standard, and other anastomotic modalities may have some impact on the postoperative observation index, which requires further refinement of subsequent studies [30]. Second, our study has a short follow-up period, which makes it difficult to find the ultimate benefit of patients; and finally, our study is a single-center study with a limited sample size, which needs to be confirmed by further multicenter prospective studies.

## 5. Conculsions

NHT can lower the difficulty of surgery, reduce positive surgical margin rate, and help recovery in short-term urinary control in patients with high-risk prostate cancer after LRP. However, we do not have evidence on the benefit of NHT in high-risk PCa patients treated with RALP. For these patients, surgery can be performed as early as possible.

**Author Contributions:** Conception, study design, writing—review and editing, G.S., S.M. and R.L.; collection of patient follow-up data, S.C. and Z.L.; data collation and verification, data analysis, Y.J. All authors have read and agreed to the published version of the manuscript.

**Funding:** This research was supported by Tianjin science and technology plan project (19ZXDBSY00050).

**Institutional Review Board Statement:** The study was conducted in accordance with the Declaration of Helsinki, and approved by Tianjin Medical University Ethics Committee (protocol code KY2021K118).

**Informed Consent Statement:** All enrolled individual participants in the study were provided with informed consent.

**Data Availability Statement:** The data presented in this study is available on request from the corresponding author.

**Conflicts of Interest:** The authors declare no conflict of interest.

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
