# Peer review of "Clinical Analysis of Perioperative Outcomes on Neoadjuvant Hormone Therapy before Laparoscopic and Robot-Assisted Surgery for Localized High-Risk Prostate Cancer in a Chinese Cohort"

_curroncol, doi:10.3390/curroncol29110683_

Round 1

Author Response

Dear Professor:

Thank you for spending time reviewing our article and for the comments made by the experts. we have read your suggestions carefully, your questions are very professional and practical, through your questions, we have further reviewed the latest literature and guidelines, revisited the purpose and data of our study, made changes to the deficiencies, and further improved the level of our study.In response to the questions, we have made the following explanations.

  1. Thank you again for your careful reading, we apologize for our oversight, we incorrectly filled in the date of inclusion of the case and made a correction.
  2. The statement NHT > 3 months was not accurate, it should be > 3 months for patients treated with NHT, we have revised this.
  3. In the LRP cohort, we found that NHT treatment contributed to the recovery of urinary control, but not in the overall cohort, which is not conflicting, and we further divided the time to recovery of urinary control into three groups and made statistical comparisons.
  4. For Table III progression-free survival and Figure 3 data are inconsistent because we used different statistical methods, but according to other expert opinions, we removed Figures 1-3.
  5. Your suggestion about the side effects of ADT treatment was very used and we described it in the discussion section and cited the relevant literature, but because our study was retrospective, cardiovascular risk stratification was not performed initially, which is to be further studied in our follow-up.
  6. For the effect of urethral anastomosis modality on urinary control, we described in the discussion section of the article, due to more factors and to keep the consistency of the cohort, we only included cases with standard modality anastomosis.

Sincerely Yours,

Guangyu Sun

The second hospital of Tianjin Medical University

Reviewer 2 Report

In the introduction the authors state that high-risk prostate cancer has no standard of care recommendation for treatment. This is not true, EAU guidelines give clear recommendations for radiotherapy with long term ADT or radical prostatectomy eventually in the ambience of a multimodal treatment. Recently the STAMPEDE trial from Attard, G et al. Showed an OS benefit for adding abiraterone for 2 yrs to 3 yrs of ADT and RT in a “Stampede” high risk definition population. Therefore the introduction has to be rewritten.

The methods section describes that patients have been staged with bone scan and MRI pelvis. This is actually not according to guidelines, a CT thorax and abdomen has to be performed to exclude metastatic disease, nowadays even if available a PPS PET, but that wasn’t accurate in the years of the retrospective enrollment. 

Author Response

Dear Professor:

Thank you for spending time reviewing our article and for the comments made by the experts. we have read your suggestions carefully, your questions are very professional and practical, through your questions, we have further reviewed the latest literature and guidelines, revisited the purpose and data of our study, made changes to the deficiencies, and further improved the level of our study.In response to the questions,we have revised the description related to treatment of high-risk prostate cancer in accordance with your comments and added the exclusion criteria to exclude distant metastases by performing thoracoabdominal CT.

Sincerely Yours,

Guangyu Sun

The second hospital of Tianjin Medical University

Reviewer 3 Report

Congratulations on your paper. It is well-written and presents important results with obvious limitations related to selection bias and retrospective nature of the study, which should be adressed in the discussion and limitation section, which should be added.

1.  According to current guideline neoadjuvant ADT should not be used outside the clinical trials. It is not widely used, but this may change in the future. Please correct the following sentence.

"Clinically, neoadjuvant hormone therapy (NHT) and the following laparoscopic RP (LRP) or robot-assisted LRP (RALP) have been widely used but the clinical findings of NHT are inconsistent among various studies"

2. Bicalutamid is an old antiandrogen. New agents such as enzalutamid, apalutamid and datolutamid are more selective and efficients. This should be discussed in the limitation section or discussion

3. Please comment on the adjuvant ADT - provide details on drugs used and rationale for that. Not every patient should receive ADT after RARP/LRP according to the current guidelines. You wrote "All patients received routine adjuvant hormone therapy after surgery."

4. Median length of hospitalization is 7 days in your study. It is higher than average. How can you explain it?

5. The rates of postive surgical margins are high in your cohort (up to 50%). Could you comment on that issue?

6. Was the type of surgery also an independent predictor of positive surgical margin? This should be mentioned and emphasized in the text

7. In the multivariate models GS at biopsy is not used properly - this is a nominal variable and should be not used as continous one. Please provide reference (e.g. GS=1) and re-calculate the model using different levels of GS (2,3,4,5). Otherwise you may categorize GS into high vs low and provide one odds ratio. I assume that not every patient had GS 4-5 at biopsy despite high-risk.

8. How can you explain that GS and T3-T4 stage are not predictors of BCR? Instead you state that prostate volume is a predictor. What about lymph nodes and extend of lymphadenectomy? Could you provide some details regarding LND and explaination for lack of significance of GS and T stage for BCR prediciton?

9. I cannot agree with the following:

"Traditionally, conservative treatments such as radiotherapy, chemotherapy, and hormone therapy are preferred for high-risk prostate cancer["

Please correct the sentence. Multimodal therapy is recommended, but this includes surgery and rather does not include chemotherapy which is reserved for metastatic PCa.

10. Please correct the sentence "For these patients, surgery can be performed as early as possible, thus avoiding unnecessary treatment". What is unnecessary treatment in this case? 

Author Response

Dear Professor:

  Thank you for spending time reviewing our article and for the comments made by the experts. we have read your suggestions carefully, your questions are very professional and practical, through your questions, we have further reviewed the latest literature and guidelines, revisited the purpose and data of our study, made changes to the deficiencies, and further improved the level of our study.In response to the questions, we have made the following explanations.

1.1 According to the expert opinion we added the limitations of the article including selection bias in the discussion section and corrected the description related to the scope of application of preoperative neoadjuvant hormone therapy for RP.

1.2 Bicalutamide is indeed a relatively old anti-androgen drug, but since our data came from three years ago, the new endocrine drugs are not widely used and the data are not comprehensive, we added the relevant description in the limitations of the article in the discussion section.

1.3 You are quite correct in pointing out the problem. We reviewed the guidelines and did not recommend routine postoperative ADT for high-risk prostate cancer, we changed the description to all patients with adverse pathology received routine adjuvant hormone therapy ± external beam radiotherapy after surgery, but in the actual clinical condition, some patients refused external radiotherapy, so only ADT treatment was given.

1.4 For the issue of hospital length of stay may have been relatively high due to the late staging and higher age of the population we included in the cohort, which included preoperative cardiopulmonary function and overall condition.

1.5 In the laparoscopic cohort, the positive margin rate for non-endocrine therapy reached 49.25%, which may be due to the relatively high surgical difficulty of performing LRP for high-risk prostate cancer due to late tumor staging and the large volume of surrounding adherent prostate; in the overall cohort, the positive margins rate was 34%, and neoadjuvant hormone therapy was beneficial in reducing the positive margins rate in the laparoscopic cohort.

1.6 We have added relevant descriptions regarding the surgical approach to the results and discussion.

1.7 Your comments are valuable. In the univariate and multifactorial logistic regression analyses, we incorrectly used GS as a continuous variable, so we have revised them, but because our study population was high-risk prostate cancer, patients with GS7 were relatively poor and statistically biased, so we divided GS into GS7-8 and GS9-10.

1.8 For the analysis of risk factors for biochemical recurrence, univariate analysis suggested that prostate volume, PSA, GS, T-stage, and positive margins were associated with biochemical recurrence, but multiple analysis only suggested that PSA and positive surgical margins were independent risk factors for biochemical recurrence, which may be due to the confounding of other factors in the univariate analysis, and may also be due to the effect of prostate volume on surgical operation and positive margins. The effect of both GS and tumor stage on biochemical recurrence was not significant in our cohort, probably due to the selectivity of our population, which included high-risk prostate cancer, mainly with a GS score of 8-10, and T3 or T4 stage was not in the overall population. The benefit of lymph node dissection and the extent of lymph node dissection for patients with high risk prostate cancer is still controversial, therefore our study did not include patients with abnormal lymph nodes, and the data collection on lymph node transitions after surgery is not perfect, which requires further study at a later stage.

1.9 We have revised and re-described our conclusions in accordance with your comments. Our study concluded that neoadjuvant hormone therapy is not necessary for patients with high-risk prostate cancer who are willing to undergo RALP therapy, and that it does not have a significant impact on perioperative outcomes and biochemical recurrence.

Sincerely Yours,

Guangyu Sun

The second hospital of Tianjin Medical University

Reviewer 4 Report

In the manuscript titled ‘Clinical analysis of perioperative outcomes on neoadjuvant hormone therapy before laparoscopic and robot-assisted surgery for localized high-risk prostate cancer in a Chinese cohort’, the authors explored the perioperative outcomes of NHT in a Chinese population. They utilized the clinical data of the patients harboring localized high-risk prostate cancer during a period of 3 years. They used data from patients with simple surgery and perioperative NHT. The major groups of the study included patients undergoing laparoscopic radical prostatectomy and robot-assisted LP. Their findings suggest that NHT can lower the difficulty of surgery as determined through measurement of operative time, blood loss, the proportion of downregulation of Gleason Score along with the recovery of urinary control among others. The manuscript is scientifically sound based on the methods and observations, however, a few improvement points are suggested for publication in the journal Current Oncology.

Critique: 

Introduction/background of the study is quite short and requires a detailed analysis of the problem statement of the study with relevant literature.

The authors claimed that ‘Currently there is no standard treatment for high-risk prostate cancer, it is recommended to revise this statement as new targeted approaches have been utilized which although are not standard but are promising. New approaches need to be acknowledged.

There is no reference to the ‘Comprehensive treatments’ statement in the introduction.

It is recommended to highlight the ‘inconsistencies among various studies’ mentioned for NHT (ref 5).

Please elaborate on the two surgical approaches further (LRP, RALP) so that a thorough background of the study can be portrayed. The authors have not clearly explained the research question of the study in the introduction.

Although it could be obvious why ‘patients with metastatic lesions’ were excluded from the study but needs to be explained in section 2.1. as to why it is out of the scope of this study.

The description of the control group in comparison to the NHT group in Table 1 is not explained in the materials and methods section, just explains the main variables and the differences such as no NHT and NHT and both groups had sub-groups of two types of interventions. 

The description of perioperative outcomes individually is also missing in the Materials and Methods section. For a broader understanding of the data and its relevance, please supply meaningful descriptions of each indicator along with the methods of measurement (for example biochemical recurrence as serum levels of PSA or the explanation of PSA progression-free survival and its difference from biochemical recurrence).

It is recommended to make sub-sections in the Results for easy comprehension of the data such as the difference between the control and NHT group data can be shown separately than for LRP and RALP difference to control.

The data in Figures 2 and 3 is already presented in Tabular form so the figure can be omitted (as the resolution of the figure is quite low and it represents no statistically significant difference).

As the authors mentioned in the Discussion, NHT is associated with complications (ref 9), how is the present study provide contradictions or support to previous data? And if there is a difference, what could be the reason according to the authors?

The authors concluded that the high accuracy of RALP reduces the impact of prostate volume and stricture space during the procedure. Please elaborate on this point further in the context of the findings of the study where perioperative outcomes of operative time, blood loss, and positive surgical margin rate were found to be not statistically different between the NHT and the control group. Is there any other similar study or data that validates this data (real-life example or case study)?

Author Response

Dear Professor:

Thank you for spending time reviewing our article and for the comments made by the experts. we have read your suggestions carefully, your questions are very professional and practical, through your questions, we have further reviewed the latest literature and guidelines, revisited the purpose and data of our study, made changes to the deficiencies, and further improved the level of our study.In response to the questions, we have made the following explanations.

1.1-1.5 Based on your comments we have added a description of the surgical methods and a comprehensive treatment for high-risk prostate cancer in the background section to further emphasize the purpose of our study, compare the current literature related to neoadjuvant hormone therapy for high-risk prostate cancer, and describe the lack of data related to new endocrine therapy in our study due to its retrospective nature and relatively long duration in the discussion section.

1.6 Our study did not include metastatic prostate cancer due to the controversy of performing surgical treatment for metastatic prostate cancer which is described in the methods section.

1.7-2.0 In the Materials and Methods section we added descriptions for the NHT and control groups and for the LRP and RALP groups. Similarly, we describe the definition of biochemical recurrence; PSA biochemical recurrence was defined using the European Urological (EAU) criteria i.e. two consecutive serum PSA levels >0.2ng/ml and Progression-free survival was defined as time from prostatectomy to In the results section, the order and description were reorganized and Figures 1-3 were deleted in accordance with your comments.

2.1 Due to the relative safety of radical prostatectomy, we found in our data collection that the incidence of intraoperative and postoperative complications including rectal injury, urinary incontinence, and hemorrhage in patients was extremely low and of no statistical value, therefore only urinary fistula was included in the study and the incidence of urinary fistula was found to be lower in the NHT group than in the control group (4.42% vs. 4.82), but not statistically different.

2.2 We further added the impact of robotic-assisted surgery on prostate cancer surgical outcomes, including the benefit on perioperative outcomes and survival, in the discussion section. For the impact of neoadjuvant hormone therapy on patients' perioperative indicators, we found that neoadjuvant therapy was beneficial for the overall cohort in terms of operative time, intraoperative bleeding and surgical margins, especially for the LRP cohort, and for the RALP cohort was not significant and is described in our discussion section.

Sincerely Yours,

Guangyu Sun

The second hospital of Tianjin Medical University

Round 2

Reviewer 1 Report

Dear Authors, 

the modifications you have made to your work have been carried out with care and meticulousness. I therefore believe it is worthy of publication. 

Author Response

Dear Professor:

Thank you again for your valuable suggestions on our work. We have benefited greatly from your comments.

Sincerely Yours,

Guangyu Sun

The second hospital of Tianjin Medical University

Reviewer 2 Report

The comments have been addressed adequately. 

Author Response

(The authors gave the same response as above.)

Reviewer 3 Report

Thank You for your answers. I still have some minor comments and suggestion about improvements that should be made.

First of all, please provide information on the lymphadenectomy performance and lymph node status (staging). Lymphadenectomy is routinely recommended for high-risk PCa during prostatectomy. Please provide the  information in the methods section on the performance of lymphadenectomy. Secondly please provide in the baseline characteristic the N staging information (at least N positive/ N0/ Nx). LNI has a crucial impact on BCR and therefore should be mentioned also in the univariable analysis or at least provide a reason for not including that.

Lastly, the below sentence from the conclusions (abstract and text) should be corrected. What do you mean by "can reduce unnecessary treatment"? I would rather say that we do not have evidence on the benefit of NHT in high-risk PCa patients treated with RALP.

"NHT is not necessary for patients with high-risk prostate cancer who are ready to undergo robot-assisted surgery and can reduce unnecessary treatment."

Author Response

Dear Professor:

Thank you again for your valuable suggestions on our work. In fact, our initial study did not include data on lymph node invasion in patients because we did not study it further considering that it was a very clear risk factor for biochemical recurrence, but this led to an incomplete study, so your suggestion was very valuable and we followed your suggestion to review our data and further refine the effect of lymph node condition on biochemical recurrence to ensure the completeness of the study. In addition, we have corrected the phrase "unnecessary treatment" as you requested. Thank you again, we have benefited greatly from your comments.

Sincerely Yours,

Guangyu Sun

The second hospital of Tianjin Medical University

Reviewer 4 Report

The manuscript looks good and can be recommended for publication in revised form.

Critique:

In the 80th line of page 2, the spelling of cohort is wrong.

Author Response

Dear Professor:

Thank you again for your valuable suggestions on our work. We have benefited greatly from your comments. We corrected the spelling mistakes, and we are sorry for our carelessness.

Sincerely Yours,

Guangyu Sun

The second hospital of Tianjin Medical University